# Potential Tumor Suppressor Role of Polo-like Kinase 5 in Cancer

**DOI:** 10.3390/cancers15225457

**Published:** 2023-11-17

**Authors:** Shengqin Su, Mary Ann Ndiaye, Glorimar Guzmán-Pérez, Rebecca Michael Baus, Wei Huang, Manish Suresh Patankar, Nihal Ahmad

**Affiliations:** 1Department of Dermatology, University of Wisconsin, Madison, WI 53705, USA; ssu42@stanford.edu (S.S.); guzmnprez@wisc.edu (G.G.-P.); 2Department of Pathology and Laboratory Medicine, University of Wisconsin, Madison, WI 53705, USA; rmbaus@wisc.edu (R.M.B.);; 3Department of Obstetrics and Gynecology, University of Wisconsin, Madison, WI 53792, USA; patankar@wisc.edu; 4William S. Middleton VA Medical Center, Madison, WI 53705, USA

**Keywords:** PLK5, cancer, polo-like kinases, tumor suppressor

## Abstract

**Simple Summary:**

Cancer is a complex disease and the underlying molecular mechanisms driving cancer initiation and progression is crucial for the development of effective approaches for cancer management. Exploring the molecular landscape of cancer can shed light on novel biomarkers, therapeutic targets, and strategies for personalized medicine. In this study, we determined the potential involvement of PLK5 in multiple cancers employing cancer tissue microarrays (TMAs). We found a downregulation of PLK5 in these cancers. Further, using publicly available GTEx and TCGA databases, we validated our findings and extended our investigations to additional cancer types. Overall, our data suggested a potential tumor suppressor role for PLK5.

**Abstract:**

The polo-like kinase (PLK) family of serine/threonine kinases contains five members (PLK1–5). Most PLKs are involved in cell cycle regulation and DNA damage response. However, PLK5 is different as it lacks a functional kinase domain and is not involved in cell cycle control. PLK5 remains the least-studied family member, and its role in oncogenesis remains enigmatic. Here, we identified tissues with high PLK5 expression by leveraging the Protein Atlas and GTEx databases with relevant literature and selected ovarian, lung, testis, endometrium, cervix, and fallopian tube tissues as candidates for further investigation. Subsequently, we performed immunohistochemical staining for PLK5 on multiple tissue microarrays followed by Vectra scanning and quantitative inForm analysis. This revealed consistently downregulated PLK5 expression in these cancers compared to normal tissues. To validate and extend our findings, we performed pan-cancer analysis of *PLK5* expression using public RNAseq databases (TCGA and GTEx). We found *PLK5* is downregulated in 18 cancer types, including our selected candidates. Interestingly, we also observed PLK5 expression remains consistently low in later stages of cancer, suggesting PLK5 may have a greater role in tumor initiation than cancer progression. Overall, our study demonstrates PLK5 downregulation in multiple cancers, highlighting its role as a tumor suppressor.

## 1. Introduction

Cancer, a complex, heterogeneous, and potentially fatal disease characterized by uncontrolled cell growth and proliferation continues to pose significant challenges in global health. Understanding the underlying molecular mechanisms driving cancer initiation, progression, and metastasis is crucial for the development of effective approaches for cancer management. Exploring the molecular landscape of cancer can shed light on novel biomarkers, therapeutic targets, and strategies for personalized medicine. In this context, the polo-like kinase (PLK) family of serine/threonine kinases, primarily shown to be key regulators of cell cycle, has emerged as a compelling area of investigation in cancer research [1,2,3,4,5,6]. PLKs are evolutionarily conserved from yeast to humans and are distinguished by two essential components: the polo-box domain (PBD), which facilitates protein interactions, and the kinase domain (KD), which regulates the catalytic activity of the protein [1,2,3,4,5,6,7]. In humans, five PLK family members have been identified (PLK1-5). Although primarily involved in cell cycle regulation, several of the family members have also been implicated in other key cellular pathways and processes, including apoptosis [8], DNA damage response [9], and cancers [9,10]. The most recently identified member of the PLK family, PLK5, was initially classified as a pseudogene [11,12]. However, recent evidence indicates that PLK5 encodes a functional protein [13,14]. Interestingly, although PLK5 is conserved in most mammals, human PLK5 has a marked difference as compared to others such as mouse. Full-length mouse PLK5 protein consists of 599 amino acids, including a complete KD and two PBDs [5,7,14]. In contrast, human PLK5 contains a premature stop codon in the KD, followed by an in-frame ATG codon [13]. Consequently, the human PLK5 protein is truncated, resulting in the translation of a smaller protein containing 336 amino acids [5,13]. Despite this incomplete KD in human PLK5, the protein appears to be functional, as evidenced by an impaired G2/M checkpoint after etoposide treatment in cells depleted of PLK5 by shRNA [14]. Curiously, while in mice the PLK5 protein has a full length KD, it has been shown to lack kinase activity, suggesting that PLK5 function may be independent of its kinase domain [7,13,15].

Interestingly, although PLK5 is generally a low expressing protein, its levels have been found to be higher in more differentiated and specialized tissues such as ovary, brain, and eye [13]. Among these tissues, PLK5 expression is particularly high in the cerebellum, suggesting its potential involvement in neuronal development and function. Nevertheless, the role PLK5 plays in different cells and tissues remains vastly unknown, despite its identification over two decades ago. To date, very limited research has been performed regarding the role of PLK5 in cancer, with only a few studies showing decreased PLK5 expression in non-small-cell lung cancer [16,17], breast cancer [18], medulloblastoma [19] and glioblastomas [13]. Additionally, a 2016 study found a novel metastasis-associated three-nucleotide deletion mutation of PLK5 in clear cell renal cell carcinoma [20] and a 2020 study identified a rare p.G223V variant of PLK5 in a small subset of ovarian cancer patients [21]. Therefore, given the limited available knowledge and potential impact of changes to PLK5 expression and/or sequence, it is crucial to expand our understanding to further elucidate the involvement of PLK5 in cancer. In this study, we employed quantitative measurement of immunohistochemical staining to conduct analysis of PLK5 protein expression on a range of disease-specific tissue microarrays (TMAs), including lung, cervix, ovarian, endometrial, fallopian, and testicular cancers. Additionally, using GTEx and TCGA databases, we validated our findings in these tumor types and extended our investigations to an additional 12 cancer types. Further, using the TCGA database, we looked at overall survival and progression-free survival of cancer patients with high and low PLK5 expression levels to determine if PLK5 expression may have an influence on survival. Overall, our data suggested a potential tumor suppressor role for PLK5.

## 2. Materials and Methods

### 2.1. Immunohistochemistry

Tissue microarrays (TMAs) were obtained from US Biomax (Rockville, MD, USA) (Cervix: #CR1001b; Endometrium: #OD-CT-RpUTR03-02; Fallopian Tube: #UTE601; Lung: #BC04002a; Normal Ovary: #OV806; Ovarian Disease: #OV1005b; Testis: #TE803) and University of Wisconsin (UW) Carbone Cancer Center Translational Science BioBank (Ovarian Cancer TMA). Immunostaining was performed by the UW Translational Research Initiatives in Pathology (TRIP) Lab on a Ventana Ultra BioMarker Platform (Roche, Indianapolis, IN, USA). Deparaffinization was carried out on the instrument, as was heat-induced epitope retrieval with CC2 buffer, a citrate-based buffer, for approximately 60 min at 100 °C. Primary antibody (Anti-PLK5 (#HPA035024; Sigma-Aldrich, St. Louis, MO, USA) diluted 1:50 in DaVinci Green (#PD900H; BioCare Medical, Pacheco, CA, USA)) was added and the slides were incubated for 60 min at 37 °C. After rinsing, discovery OmniMap anti-Rabbit HRP was added (Ventana #760-4311; Roche) and incubated for 16 min at 37 °C before rinsing again and adding Discovery ChromoMap DAB detection (Ventana #760-159; Roche). The slides were then rinsed again and Cytokeratin primary antibody (Dako #M3515; Agilent, Santa Clara, CA, USA) diluted 1:1000 in DaVinci Green (BioCare Medical #PD904H) was applied and incubated for 8 min at 37 °C. This was followed by incubation with Discovery UltraMap anti-Mouse AP (Ventana #760-4312; Roche) for 8 min at 37 °C and subsequent detection with Discovery RED (Ventana #760-228; Roche). The slides were counterstained with hematoxylin (1:5) for 30 sec, rinsed, dehydrated and coverslips were mounted.

### 2.2. Vectra Scanning/inForm Analysis

TMA slide scanning and image acquisition was carried out using the Vectra automated multispectral imaging system (Akoya Biosciences, Marlborough, MA, USA). Employing the Nuance software (version 3.02; PerkinElmer, Waltham, MA, USA) we created a spectral library in order to unmix the signals on the acquired images stained with DAB, Discovery RED, and hematoxylin. This aids in the recognition of the different chromogens based on their unique spectral curves and to analyze protein levels in each TMA core. Using the inForm software (version 2.4.8; Akoya Biosciences, Marlborough, MA, USA), we trained the program on distinguishing between different tissue segments (epithelium vs. stroma) and cell segments (nucleus, cytoplasm, cell membrane) within the TMA tissue cores. Following the software training, quantitative image analysis was carried out in each tissue core using inForm.

### 2.3. Statistical Analysis for Tissue Microarrays (TMAs)

In each of the tissue cores, values for epithelium only were used. The normalized optical density (OD) per unit area [OD/peak weighting value within each cell was divided by the cell area (square microns)] for the DAB chromogen was considered as the level of PLK5 expression. Statistical analysis was conducted using R software (v4.3.1) and its developmental environment RStudio (v 2023.06.1+524). PLK5 expression was compared across the disease stages in different cancer types. The boxplot between two or multiple groups and simple linear regression were conducted using ggplot2 (v3.3.3) and ggpmisc (v0.5.2) packages.

### 2.4. TCGA and GTEx Database Analyses of PLK5 Expression

The results shown here are in part based upon data generated by The Cancer Genome Atlas (TCGA) research network: https://www.cancer.gov/tcga (accessed on 10 May 2023). The Genotype-Tissue Expression (GTEx) project was supported by the common fund of the Office of the Director of the National Institutes of Health and by NCI NHGRI NHLBI NIDA NIMH and NINDS. The data used for the analyses described in this manuscript were obtained from GTEx Analysis Release V8 (dbGaP Accession phs000424.v8.p2).” *PLK5* expression profiles in normal tissues was queried in the GTEx portal (https://www.gtexportal.org/home/gene/PLK5 (accessed on 10 May 2023)). *PLK5* expression between normal vs. tumor in selected tissues were downloaded from OncoDB [22]. The pan-cancer and stage-wise analysis of *PLK5* were queried in Gepia2 [23]. T-test was performed with q < 0.05 as significance cutoff for pan-cancer *PLK5* analysis. ANOVA was conducted to compare *PLK5* stage-wise expression analysis in selected cancers. The correlation of *PLK5* and age/sex were queried in cBioPortal [24]. For TCGA/GTEx analysis, *PLK5* expression is plotted in log2(TPM+1) scale.

### 2.5. Survival Analysis of PLK5 Expression in TCGA Database

The available data on PLK5 expression and the overall and progression-free survival of patients were downloaded from cBioPortal for selected cancer types (accessed on 6 November 2023) [25]. Subsequently, we conducted Cox proportional-hazards (PH) regression and log-rank test using the survival (v3.5-7) and survminer (v0.4.9) packages in R. For Cox proportional-hazards (PH) regression, we computed the hazard ratio and the *p*-value from the Wald test to assess the association between PLK5 expression and the survival of cancer patients.

For the log-rank test, the cancer patients were divided into two groups based on their PLK5 expression: (1) If more than half of the patients had zero PLK5 expression, we categorized patients with non-zero PLK5 expression as “High PLK5,” while those with zero PLK5 expression were labeled as “Low PLK5.” (2) If more than half of the patients had non-zero PLK5 expression, we assigned patients with PLK5 expression greater than or equal to the median as “High PLK5,” and those with PLK5 expression less than the median as “Low PLK5.” Next, Kaplan–Meier survival curves were generated for the two groups using survminer (v0.4.9) within R.

## 3. Results and Discussion

### 3.1. PLK5 Expression Is High in the Normal Tissues of Brain, Respiratory System, and Reproductive System

To determine the tissue-specific expression pattern of PLK5 and its potential association with cancer, we conducted a comprehensive analysis using publicly available databases. First, we utilized data compiled by the Human Protein Atlas [26,27,28,29] on PLK5 expression across 44 different normal tissue types as determined by immunohistochemical staining and interpretation by trained pathologists. PLK5 expression was shown to be highest in nasopharynx, bronchus, testis, and fallopian tube tissues, while the remaining tissues exhibited very low PLK5 (Figure 1A).

To validate these findings, we analyzed the mRNA expression profile of *PLK5* using the GTEx database. We found that while *PLK5* expression is generally low across most tissues, it is significantly elevated in brain tissues (cerebellum, cortex, and pituitary), female reproductive tissues (cervix, fallopian tube, uterus, vagina, ovary, and breast), and male reproductive tissues (testis and prostate) (Figure 1B). To reinforce the robustness of our findings, we performed additional analysis using two RNAseq databases, HPA (Human Protein Atlas [28]) and FANTOM5 [30,31]. The results from these independent cohorts are consistent with the findings that PLK5 is highly expressed in brain as well as female and male reproductive tissues (Appendix A).

### 3.2. PLK5 Expression Is Downregulated in Cancer Tissues, as Shown by TMA Analyses

In view of earlier published studies supporting a tumor suppressor role of PLK5 expression in non-small-cell lung cancer [16,17], breast cancer [18], medulloblastoma [19] and glioblastomas [13], we were interested in expanding the available knowledge on the role of this kinase in additional cancer types. Our observation of high PLK5 expression in certain specific normal tissues led us to further investigate the expression profile of PLK5 in cancers of these same tissue types to determine its potential tumor suppressor role in these tissues. For this purpose, we employed multiple tissue microarrays (TMAs, for more information please see Appendix A) containing samples from lung, cervix, ovary, fallopian tube, endometrium, and testis (Figure 2A). The TMAs were immunostained for PLK5, followed by Vectra scanning and inForm analysis for quantitative analyses of protein expression, where we compared the PLK5 protein levels in cancer tissues versus their normal counterparts (Appendix A). We excluded brain tissue from our investigation, due to an earlier published paper on certain brain cancers [13]. As shown by our data, we found significantly reduced levels of PLK5 protein in lung, cervical, ovarian, and fallopian tube cancers, and a decreasing trend towards significance in endometrial and testicular cancer tissues (Figure 2B). These findings are consistent with a potential tumor suppressor role of PLK5 as reported earlier in certain cancers as mentioned earlier. However, additional detailed studies are required to determine the exact role of PLK5 in cancer.

### 3.3. PLK5 Expression Is Reduced in Multiple Malignant Tissues Compared to Normal Tissues, as Shown by Analysis of the GTEx and TCGA Databases

To validate our findings from the TMA analysis, we corroborated our results using independent cohorts from the GTEx and TCGA databases. We compared the *PLK5* expressions in normal tissue from both TCGA and GTEx alongside cancer tissues from TCGA. Encouragingly, our results were shown to be aligned with our earlier TMA analysis findings, and we found a consistent downregulation of *PLK5* expression in ovary (OV; ovarian serous cystadenocarcinoma), cervical (CESC; cervical squamous cell carcinoma and endocervical adenocarcinoma), endometrial (UCEC; uterine corpus endometrial carcinoma), and testicular cancers (TGCT; uterine corpus endometrial carcinoma) (Figure 3A). This congruence between TMA analysis and the independent GTEx/TCGA datasets lends robustness to our findings and strengthens the reliability of our conclusions regarding a potential tumor suppressor function of PLK5.

To broaden the scope of our investigation, we conducted a pan-cancer analysis of *PLK5* expression using the extensive TCGA and GTEx databases (Figure 3B). As shown in the figure, we observed a broader downregulation of *PLK5* in an additional 13 malignancies arising from tissues with both high and low *PLK5* expression. Notably, we confirmed a reduction of *PLK5* in cancers originating from tissues with elevated *PLK5* expression, including brain cancers (e.g., glioblastoma, low-grade gliomas), breast cancers, prostate adenocarcinoma, and tenosynovial giant cell tumors. Moreover, reduced *PLK5* levels were also observed in cancers arising from tissues with relatively low *PLK5* expression, including colon adenocarcinoma, diffuse large B-cell lymphoma, esophageal carcinoma, kidney chromophobe, acute myeloid leukemia, rectum adenocarcinoma, cutaneous melanoma, thyroid carcinoma, thymoma, and uterine carcinosarcoma. Our findings suggest that *PLK5* may play an important role in cancer and may have potential as a diagnostic or therapeutic biomarker.

### 3.4. PLK5 Levels Remain Consistent during Cancer Progression and Are Independent of Age or Sex

To further investigate the involvement of PLK5 in cancer progression and metastasis, we analyzed its expression levels in various cancer stages. In our TMA dataset, we observed that, although PLK5 levels were reduced in cancer tissues compared to normal, the PLK5 protein levels remained consistently low as cancer progressed (Figure 4A). Analysis of *PLK5* levels across different cancer stages using TCGA analysis corroborated our results, demonstrating that during the progression of cancer, *PLK5* expression remained stable in cervix, endometrium, lung, ovary, and testicular cancers (Figure 4B). Additionally, our analysis of an additional nine tumor types showed the same trends of stable *PLK5* expression after the initial reduction in expression (Appendix A). Collectively, these findings suggest that although PLK5 may be involved in tumor initiation and/or early-stage development, its significance may be limited in cancer progression and metastasis once a malignancy is established. Further experiments are warranted to fully elucidate the precise role of PLK5 in carcinogenesis and cancer progression.

We next sought to explore the correlation of PLK5 expression with age and/or sex in the context of our TMAs as well as TCGA database (Appendix A). In our TMA analysis, we found decreased PLK5 expression with increasing age in fallopian and cervical cancer, as well as increasing PLK5 with increasing age in ovarian cancer (Appendix A). However, no significant correlation was observed between PLK5 expression and age in endometrium, lung, or cervical cancer. On the other hand, other than lung adenocarcinoma (LUAD), no significant correlation was found between age and *PLK5* expression in any of the other cancers in the TCGA database, although a trend was seen in the testicular cancer dataset. Similarly, no significant correlation between PLK5 and sex was found in lung cancer in either the TMA or TCGA datasets (Appendix A). These findings highlight the potentially complex nature of PLK5 regulation in different cancer types. Further studies are warranted to unravel the intricate interplay between PLK5 expression, age, sex, and specific cancer types to enhance our understanding of its clinical implications.

### 3.5. PLK5 Levels Have No Association with the Overall or Progression-Free Survival in Various Cancer Types

Given that PLK5 expression remains consistently low during cancer progression, we anticipate that PLK5 plays a limited role in cancer prognosis. To further investigate this, we conducted Cox proportional-hazards (PH) regression analysis on cervix, endometrium, lung, ovary, and testis cancers using the TCGA database. Our analysis revealed that the hazard ratio for PLK5 is not significantly different from 1 in both overall survival (Figure 5A) and progression-free survival (Figure 5B). This indicates that PLK5 expression does not appear to impact cancer prognosis in these cancers. In addition to these findings, Kaplan–Meier curves were plotted to compare survival between PLK5-high and PLK5-low patients. Similar to the hazard ratio, the data show no discernible difference in overall survival (Figure 5C) or progression-free survival (Figure 5D) among most cancer types. Furthermore, we extended our hazard ratio and Kaplan–Meier analyses to an additional nine tumor types we identified earlier. Similar to the other cancers tested, these results showed no consistent association between PLK5 expression and patient survival (Appendix A). In summary, our findings suggest that PLK5 expression remains low during cancer progression and has negligible influence on patient survival. This is not an unexpected finding since PLK5 levels, in general, are extremely low in cancer throughout the different stages of cancer progression. However, although there appears to be no effect on survival, further research is needed to determine what role PLK5 may play in cancer due to its reduced expression in cancer tissues and the tumor suppressive qualities found by us and others.

## 4. Conclusions

The role of PLK5 in cancer remains vastly understudied, warranting further research to clarify its role in cancer development and progression. In this study, we have found that PLK5 is downregulated in multiple types of cancers, suggesting a potential tumor suppressor function of this protein. Interestingly, we observed that PLK5 expression remains consistent as cancer progresses and that PLK5 expression does not affect either overall survival or progression-free survival in the cancer types we tested, suggesting a limited role for PLK5 in driving cancer progression and metastasis. However, in-depth research is necessary to confirm these results in independent datasets and experimental models.

While the downregulation of PLK5 in many cancers holds promise for further research, the mechanisms underlying its regulation and potential therapeutic implications are still elusive. Future studies could explore the molecular pathways involved in PLK5 regulation and its interactions with other PLK family members and cancer-related genes, which may provide valuable insights into its contribution to tumorigenesis. A detailed understanding of the upstream and downstream effectors of PLK5 could be very useful towards the potential clinical exploitation of this target in cancer. Additionally, further investigation is needed to determine if PLK5 could be used as a target for cancer management. Continued research in these areas will advance our understanding of the complex role of PLK5 in cancer and may unveil novel avenues for cancer management. For example, novel small molecule and natural product agonists could be developed for cancer chemoprevention and cancer interception, since PLK5 appears to be involved in the early stages of the carcinogenesis process.

## Figures and Tables

**Figure 1 cancers-15-05457-f001:**
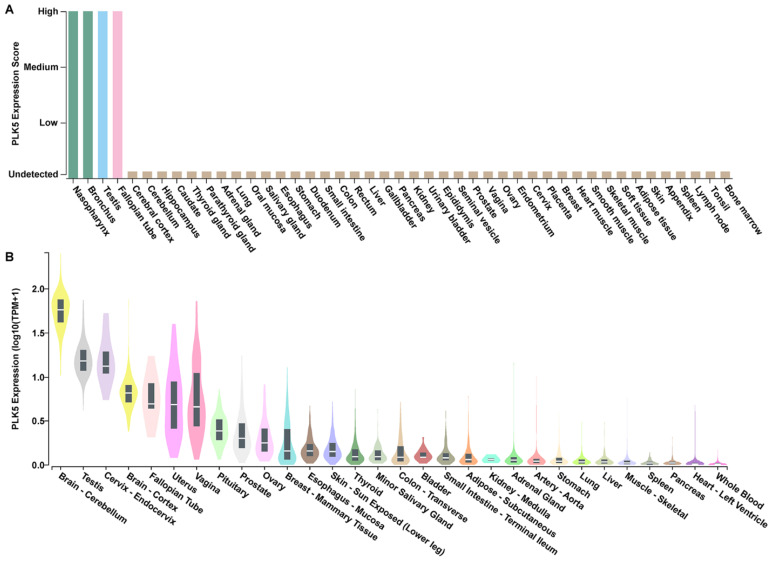
PLK5 expression using publicly available databases. (**A**) PLK5 protein expression profile using immunohistochemical staining of pan-tissue TMAs containing 44 different normal tissue types, as available from proteinatlas.org. Image credit: Human Protein Atlas, www.proteinatlas.org (accessed on 10 May 2023) [28]. Image has been edited for style only and original image is available at the following URL: v23.proteinatlas.org/ENSG00000185988-PLK5/tissue. (**B**) *PLK5* mRNA expression as determined using the GTEx RNA-Seq database consisting of 54 non-diseased tissue sites from ~1000 individuals to create a public dataset. Tissues with the same organ of origin are labeled with the same color.

**Figure 2 cancers-15-05457-f002:**
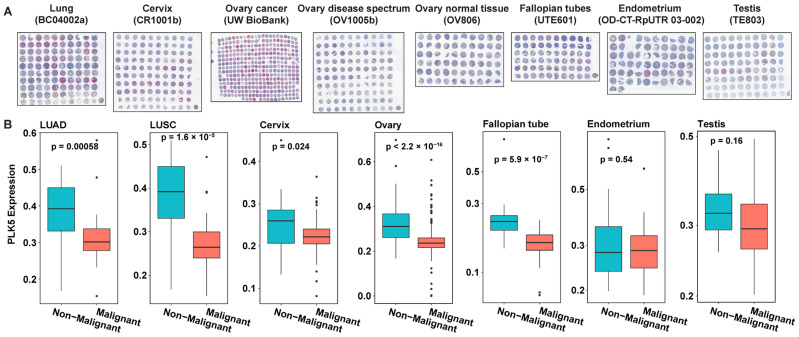
PLK5 protein levels in multiple cancer tissue microarrays (TMAs). (**A**) Composite staining images of TMAs used in the analysis. PLK5 is stained with DAB (brown), cytokeratin (pink) is used as a tumor marker, and blue is a nuclear stain. (**B**) PLK5 levels in each cancer type with differential expression shown between non-malignant and malignant tissues from each TMA. LUAD: Lung adenocarcinoma; LUSC: Lung squamous cell carcinoma.

**Figure 3 cancers-15-05457-f003:**
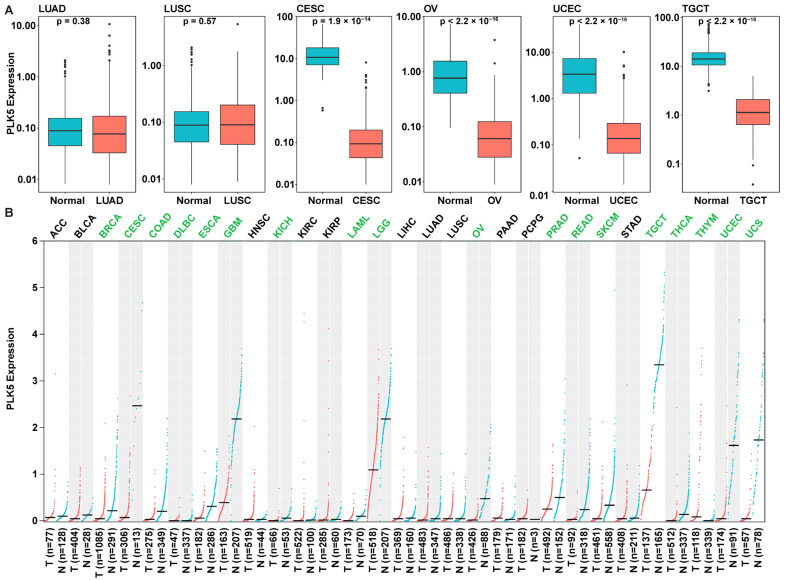
*PLK5* expression in normal tissues and cancers using GTEx and TCGA databases. (**A**) Expression of *PLK5* in same tissues as were analyzed in TMAs. (**B**) Pan-cancer analysis of *PLK5* expression. Green text means PLK5 is down regulated in tumors (q < 0.05), the black means PLK5 is not down regulated in tumors. Blue lines and boxes are normal tissues and red are cancer tissues. ACC: Adrenocortical carcinoma; BLCA: Bladder urothelial carcinoma; BRCA: Breast invasive carcinoma; CESC: Cervical squamous cell carcinoma and endocervical adenocarcinoma; COAD: Colon adenocarcinoma; DLBC: Lymphoid neoplasm diffuse large B-cell lymphoma; ESCA: Esophageal carcinoma; GBM: Glioblastoma multiforme; HNSC: Head and neck squamous cell carcinoma; KICH: Kidney chromophobe; KIRC: Kidney renal clear cell carcinoma; KIRP: Kidney renal papillary cell carcinoma; LAML: Acute myeloid leukemia; LGG: Brain lower grade glioma; LIHC: Liver hepatocellular carcinoma; LUAD: Lung adenocarcinoma; LUSC: Lung squamous cell carcinoma; OV: Ovarian serous cystadenocarcinoma; PAAD: Pancreatic adenocarcinoma; PCPG: Pheochromocytoma and paraganglioma; PRAD: Prostate adenocarcinoma; READ: Rectum adenocarcinoma; SKCM: Skin cutaneous melanoma; STAD: Stomach adenocarcinoma; TGCT: Testicular germ cell tumors; THCA: Thyroid carcinoma; THYM: Thymoma; UCEC: Uterine corpus endometrial carcinoma; UCS: Uterine carcinosarcoma.

**Figure 4 cancers-15-05457-f004:**
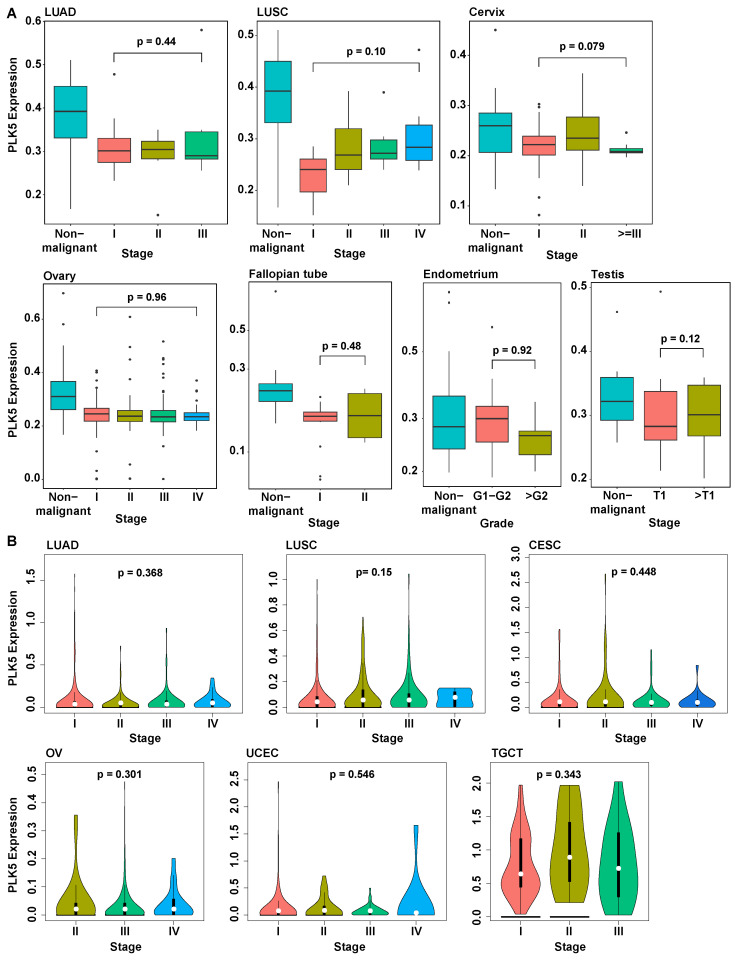
Analysis of stage-wise PLK5 expression. (**A**) Analysis of PLK5 protein levels in TMAs by tumor stage. (**B**) Analysis of TCGA data for *PLK5* expression by tumor stages of the same tumor types as in TMA. Note that TCGA does not have data for fallopian tube tumors, and therefore, it was excluded. LUAD: Lung adenocarcinoma; LUSC: Lung squamous cell carcinoma; CESC: Cervical squamous cell carcinoma and endocervical adenocarcinoma; OV: Ovarian serous cystadenocarcinoma; TGCT: Testicular germ cell tumors; UCEC: Uterine corpus endometrial carcinoma.

**Figure 5 cancers-15-05457-f005:**
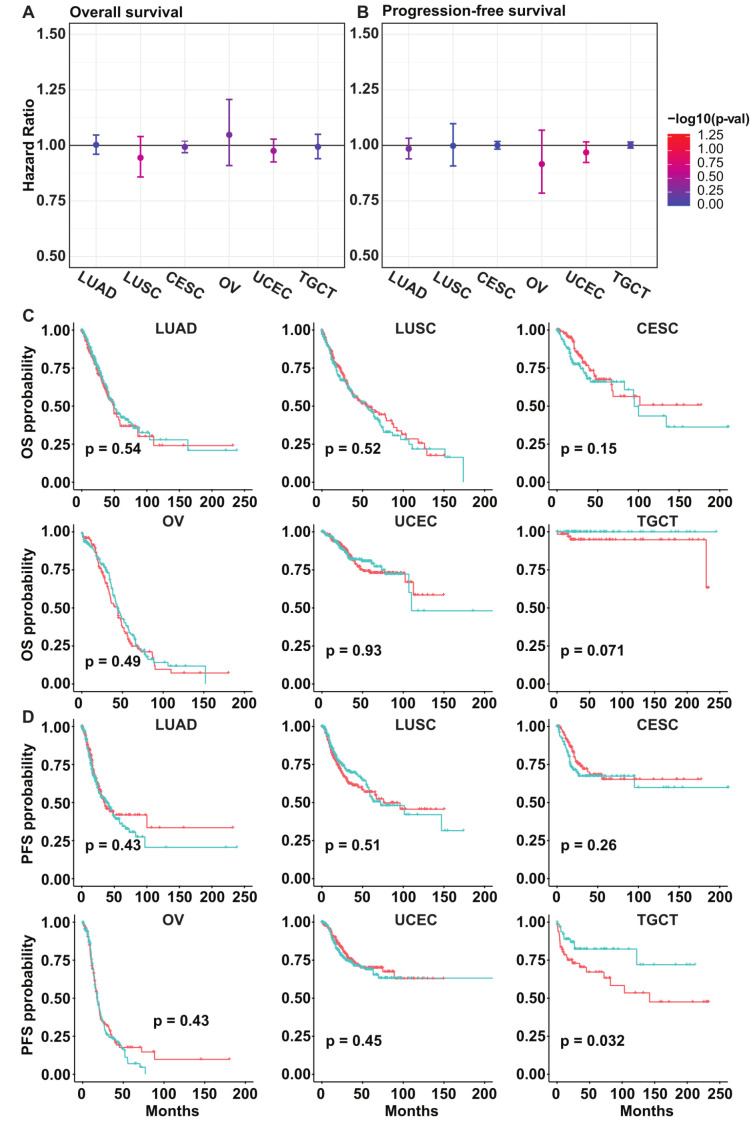
Survival analysis of cancers according to *PLK5* expression using the TCGA database. (**A**) *PLK5* hazard ratio plot for overall survival in the same tissues as were analyzed in TMAs. The error bar denotes 95% confidence interval. (**B**) *PLK5* hazard ratio plot for progression-free survival in same tissues as were analyzed in TMAs. The error bar denotes 95% confidence interval. (**C**) Kaplan–Meier plots comparing the overall survival between PLK5-High and -Low patients. (**D**) Kaplan–Meier plots comparing progression-free survival curves between PLK5-High and -Low patients. For (**C**,**D**), red color is PLK-High patients, while blue color is PLK-Low patients. CESC: Cervical squamous cell carcinoma and endocervical adenocarcinoma; LUAD: Lung adenocarcinoma; LUSC: Lung squamous cell carcinoma; OV: Ovarian serous cystadenocarcinoma; TGCT: Testicular germ cell tumors; UCEC: Uterine corpus endometrial carcinoma.

## Data Availability

Data used are from publicly available databases and are accessible to everyone. Our tissue microarray data has been provided in the Appendix A section as well.

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
