# Peer review of "Potential Tumor Suppressor Role of Polo-like Kinase 5 in Cancer"

_cancers, 2023, doi:10.3390/cancers15225457_

Round 1

Reviewer 1 Report

Comments and Suggestions for Authors

The manuscript "Potential tumor suppressor role of Polo-Like Kinase 5 in cancer" by Shengqin Su et al. establishes a pattern of downregulation of Polo-Like Kinase 5 (PLK5) in a broad array of human malignancies. This is an important study showing potential tumor suppressor role of PLK5 in cancer, especially given poorly understood function of PLK5 and limited published data on its role in tumorigenesis. Interestingly, the authors also shown that PLK5 expression remains consistent as cancer progresses, suggesting a limited role for PLK5 in driving cancer progression and metastasis.  This is a very well-done study that provides new insights into the role of PLK5 in cancer. The experiments are logical and the conclusions are consistent with the data. The manuscript is very well written and presented.

However, the following minor criticism needs to be considered:

11.     Figure 2A would be more informative if shown at higher magnification as a supplemental figure.

22.     The observations that PLK5 protein levels remained consistently low as cancer progressed is very interesting: do authors therefore predict that loss of PLK5 in tumors is not associated with worse clinical outcomes? Are there data to support this hypothesis?

33.     Since PLK5 lacks a functional kinase domain, to avoid confusion, please replace "kinase" with "protein" in line 263.

Author Response

Comment/Concern: The manuscript "Potential tumor suppressor role of Polo-Like Kinase 5 in cancer" by Shengqin Su et al. establishes a pattern of downregulation of Polo-Like Kinase 5 (PLK5) in a broad array of human malignancies. … The manuscript is very well written and presented.

Response: The authors would like to thank Reviewer #1 for their careful reading of our manuscript, encouraging comments, and suggestions for improvement. We have addressed the concerns of the Reviewer below.

Comment/Concern: Figure 2A would be more informative if shown at higher magnification as a supplemental figure.

Response: The TMA images are truncated mosaics of multiple images (~4 20x images per core). Due to this, when magnified, they appear very pixelated and do not add additional information. We included them in Figure 2 as a representation to the readers of the TMAs we stained. Additionally, PLK5 is a very low abundance protein and is difficult to detect by eye when doing multicolor chromogenic staining, which is why we have used the Vectra/inForm method of quantification.

Comment/Concern: The observations that PLK5 protein levels remained consistently low as cancer progressed is very interesting: do authors therefore predict that loss of PLK5 in tumors is not associated with worse clinical outcomes? Are there data to support this hypothesis?

Response: In the revised manuscript, we have provided overall and progression-free survival analysis on PLK5 expression using the TCGA database. Our results showed that PLK5 expression is not associated with worse clinical outcomes. As discussed in the text, it appears that PLK5 may have a greater role in tumor initiation than cancer progression. We have updated the survival analysis to the method section in page 8, the result section in page 17-19, and the discussion section in page 20. We also created new figures as Figure 5 and Figure S5.

Comment/Concern: Since PLK5 lacks a functional kinase domain, to avoid confusion, please replace "kinase" with "protein" in line 263.

Response: We have made the suggested replacement.

Reviewer 2 Report

Comments and Suggestions for Authors

Overall, the authors used computational approaches to analyze the expression levels of PLK5, a not well studied protein kinase, in normal and malignant tissues spanning various organs, diverse cancer stages, and patient age groups.  The analysis provides valuable insights into the potential role of PLK5 in cancer formation and progression.  While the manuscript is well-written, the authors could further improve it by discussing the biological significance of their findings, especially in light of some statistically insignificant results.

Author Response

Comment/Concern: Overall, the authors used computational approaches to analyze the expression levels of PLK5…  While the manuscript is well-written, the authors could further improve it by discussing the biological significance of their findings, especially in light of some statistically insignificant results.

Response: We would like to thank Reviewer #2 for their thoughtful reading of our manuscript and suggestions. We have included extra discussion of the biological significance of our findings in the Conclusion section on pages 20.

Reviewer 3 Report

Comments and Suggestions for Authors

The study in question undertakes a comprehensive investigation into the function of Polo-Like Kinase 5 (PLK5) across diverse cancer types, specifically as a tumor suppressor. Employing a multifaceted approach, the researchers conducted immunohistochemical staining on tissue microarrays to examine the expression of PLK5 in various cancer tissues in contrast to their normal counterparts. Additionally, the study harnesses publicly available RNAseq databases, including TCGA and GTEx, to validate and extend the findings across a spectrum of cancer types.

The main findings of this research are noteworthy and can be summarized as follows:

1. PLK5 exhibits consistent downregulation in multiple cancer types compared to normal tissues.

2. The persistence of low PLK5 expression in later cancer stages suggests a more significant role in tumor initiation than in cancer progression.

3. The pan-cancer analysis, utilizing public RNAseq databases, corroborates the downregulation of PLK5 across 18 distinct cancer types.

4. The study posits that PLK5 may serve as a tumor suppressor.

However, the study has a single shortcoming that might be considered. Notably, the article lacks mention of clinical data, such as patient outcomes or survival rates correlating with PLK5 expression. I suggest to add relevant data addressing this aspect to the Results section. 

In conclusion, this study confirmed the downregulation of PLK5 in diverse cancer types, underscoring its potential role as a tumor suppressor. The identification of PLK5 as a potential biomarker and therapeutic target opens up avenues for the development of novel strategies in cancer management. The findings contribute considerably to the understanding of PLK5 role in cancer and its potential implications for clinical applications, making the study suitable for publication.

Comments on the Quality of English Language

The quality of English is good.

Author Response

Comment/Concern: The study in question undertakes a comprehensive investigation into the function of Polo-Like Kinase 5 (PLK5) across diverse cancer types, specifically as a tumor suppressor... The findings contribute considerably to the understanding of PLK5 role in cancer and its potential implications for clinical applications, making the study suitable for publication.

Response: The authors would like to thank Reviewer #3 for their thorough reading and concise breakdown of the manuscript.

Comment/Concern: However, the study has a single shortcoming that might be considered. Notably, the article lacks mention of clinical data, such as patient outcomes or survival rates correlating with PLK5 expression. I suggest to add relevant data addressing this aspect to the Results section.

Response: In the revised manuscript, we have provided overall and progression-free survival analysis on PLK5 expression using the TCGA database. Our results showed that PLK5 expression is not associated with worse clinical outcomes. As discussed in the text, it appears that PLK5 may have a greater role in tumor initiation than cancer progression. We have updated the survival analysis to the method section in page 8, the result section in page 17-19, and the discussion section in page 20. We also created new figures as Figure 5 and Figure S5.

Comment/Concern: Minor editing of English language required

Response: We have carefully read through the manuscript and edited the text (yellow highlights) in places for clarity.